# Status and subjective well-being: A conceptual replication and extension of Anderson et al. (2012)

Cameron Anderson[1], John Angus D. Hildreth[2]*

1 Walter A. Haas School of Business, University of California, Berkeley, Berkeley, California, United States of America, 2 The Samuel Curtis Johnson Graduate School of Management, Cornell SC Johnson College of Business, Cornell University, Ithaca, New York, United States of America

* hildreth@cornell.edu

**Data Availability Statement:** All data files are available from the OSF database at the following link: https://osf.io/wkr5p/?view_only=09d16afc34f44f5996d64edf2bb9fd87.

## Abstract

Does the status people possess shape their subjective well-being (SWB)? Prior research that has addressed this question has been correlational. Therefore, in the current research, we provide causal evidence of this effect: In two experiments, we found that individuals reported higher SWB when their own status was higher compared to when it was lower. However, individuals' SWB was not only shaped by their own status, but also by others' status. Specifically, individuals reported higher SWB when others' status was lower than when it was higher. Thus, people have a competitive orientation towards status; they not only want to have high status on an absolute level (e.g., to be highly respected and admired), but also to have higher status than others (e.g., to be more respected and admired than others). A standard self-affirmation manipulation was used in an attempt to mitigate individuals' competitive orientation towards status, but only helped already high-status members feel happier in groups of high-status members, rather than help low-status members feel happier when they uniquely held low status.

## Introduction

Does the status people possess shape their subjective well-being (SWB)–that is, people's cognitive and affective evaluations of their lives [1]? Status is the respect, admiration, and voluntary deference individuals are afforded by others [2]. In a 2012 paper, Anderson and colleagues [3] hypothesized that status influences SWB because it affects the power and influence people possess in their interpersonal relationships [4] and shapes their sense of belonging and social acceptance [5]–variables that strongly affect SWB [e.g., 6, 7]. In addition, status might shape SWB because it affects their self-esteem; many studies have found a link between status and self-esteem [e.g., 8], and self-esteem is also tightly associated with SWB [9].

In support of this hypothesis, multiple studies have found a correlation between status and SWB. For example, an investigation of cohorts of students found that individuals ranked more highly as "leaders" by classmates were rated as more cheerful [10]. A large cross-cultural study found that around the world, people's happiness was associated with the degree to which they

**Funding:** The author(s) received no specific funding for this work.

**Competing interests:** The authors have declared that no competing interests exist.

felt respected by others and proud of something [11]. An examination of diverse kinds of social groups found that individuals rated as higher in status by fellow group members self-reported higher levels of SWB [3]. Other variables related to status, such as employment status, also predict SWB [e.g., 12].

Because the above evidence is correlational, it does not establish the causal effects of status. It is thus possible that the observed association between status and SWB was due to third variables or arose because SWB influences status rather than vice-versa. For example, individuals who have higher subjective well-being and who tend to feel happier might attain higher status [13].

The ambiguity in causal origins makes the use of experimental designs particularly important. To our knowledge, only one study used experimental methods to establish the causal effects of status on SWB. In Study 3 of Anderson et al. (2012), participants were shown a ladder with 10 rungs that represented the level of status people have in their important social groups. Depending on the condition, they were asked to compare themselves to the people either at the top or the bottom rung of the ladder, and to think of how the similarities and differences between them and the comparison target would affect a getting-acquainted interaction. In that study, participants in the high-status condition, who were asked to compare themselves to people at the bottom rung of the ladder, reported higher SWB than did participants in the low-status condition, who compared themselves to people at the top rung of the ladder.

However, that finding did not hold up in a subsequent replication [14]. Specifically, in an aggregate replication sample that included multiple laboratories, participants assigned to the high-status condition had slightly lower SWB, rather than higher, than those assigned to the low-status condition. It is possible that this different effect was due to the specific features of the manipulation. In the high-status condition, participants were asked to think about someone with low status, and in the low-status condition, participants were asked to think about someone with high status. Although the original reasoning behind this manipulation was that it would leverage dominance complementarity effects [e.g., 15], it is possible the manipulation might have inadvertently cognitively primed the opposite mindset than what was intended. By thinking about someone with high status, participants might have been placed in a high-status mindset.

The current research had three primary aims. First, we conducted a conceptual replication of Study 3 in Anderson et al. (2012), testing whether status does causally influence SWB, but using a different status manipulation than that used in the original study. Specifically, we gave participants false feedback about their status within a social group [e.g., 16–18]–a manipulation that has consistently been shown to successfully manipulate individuals' sense of their own status vis-à-vis others. We expected status to causally affect SWB.

Second, in addition to examining the effect of individuals' own status on their SWB, we examined the effect of others' status on individuals' SWB. According to many theories, the human desire for status is inherently competitive. People not only desire to receive a high level of respect, admiration, and deference from others–they desire to receive a *higher* level of respect, admiration, and deference *than others receive* [19–21]. By definition, this *status-is-competitive* account suggests that people will be happier when others possess low status, because that means their own relative position is higher.

However, there is surprisingly little research that directly addresses this claim. In fact, some research on self-esteem suggests that people do not hold a competitive orientation toward status within their groups. Specifically, Tyler and Blader [22] measured people's self-perceptions of status on an absolute level (e.g., "Most members of [my group] respect me") as well as their status relative to others (e.g., "My ideas get more attention than those of others"). Their findings suggested people's self-esteem is shaped by their status on an absolute level but very little by their status on a relative level–that is, people seemed to have a non-competitive orientation

towards status. As they concluded, "it appears that people focus primarily upon whether or not they are members 'in good standing' and not on whether they are in 'better standing' than other group members" [22, p. 830]. However, it is unclear whether their findings, which focused on self-esteem, would hold up for SWB.

Third, if status does have a causal effect on SWB, we sought to explore whether it is possible to mitigate its effects. Status hierarchies appear to emerge in all social groups, which means it is inevitable that some group members have higher status and others have lower status [23, 24]. If individuals' status influences their SWB, however, this implies that some group members–namely those lower in status–will inevitably experience lower SWB. If that is the case, it is important to explore possible interventions or ways to mitigate the potential psychological damage of possessing lower status. To examine this question, we focused on self-affirmation, an exercise that focuses individuals' attention on the values that are most important to them [25, 26]. In a prior study, participants were less interested in boosting their own status after they engaged in self-affirmation [27], which suggests self-affirmation might mitigate the effects of status on SWB because it makes status less important to the individual.

In sum, the current research aimed to conceptually replicate Anderson et al. (2012) and test whether status has a causal effect on SWB using experimental methods, test whether individuals' SWB is affected by others' status in addition to their own, and test whether self-affirmation mitigates the impact of status on SWB. We pursued these aims in two studies. In Study 1, we placed people into small groups, provided each of them with (false) feedback about their own and others' status in the group, and assessed their SWB. Study 2 was preregistered (https://aspredicted.org/LK8_DTB). It used the same design as in Study 1, but also included a self-affirmation manipulation in which some participants were randomly assigned to self-affirm before they received their status feedback. Data and protocols for both studies are posted to OSF (https://osf.io/wkr5p/?view_only=09d16afc34f44f5996d64edf2bb9fd87).

## Study 1

### Methods

**Participants.** Participants were 230 undergraduate students from a West Coast university (41% male, 59% female; average age = 21.10 years, $SD$ = 2.79) who were paid $15 or course credit. The sample size was determined by the size of the available pool of participants which was limited to undergraduate students. No other inclusion criteria were used. Participants were recruited between April 16, 2015, and October 2, 2015. The research in both studies was conducted in accordance with the principles expressed in the Declaration of Helsinki and the protocol was approved by The Committee for Protection of Human Subjects comprising the Institutional Review Boards of the University of California, Berkeley (IRB Approval ID: 2015-01-7044). All subjects in both studies gave their informed consent for inclusion before they participated in the study and were then debriefed at the end of the study. The study was completed when the available pool of participants ran out. Participants were asked to select all racial-ethnic categories to which they belonged; 66% selected Asian American, 20% selected White, 8% selected Latino, 2% selected African American, 1% selected Native American, and 11% selected "other." During the initial questionnaire, described below, two attention check questions were included that asked participants to select specific responses, i.e., "If I am paying attention to this study, I will select the agree a little option," and "If I am paying attention I will select this answer. /I will not select this answer." This method of detecting inattentive participants has been used extensively in past research to increase the quality of data [e.g., 28]. Four participants failed attention check questions or had technical issues and were thus excluded from the analyses, leaving 226 participants.

**Procedure and status manipulation.**  The status manipulation was based on prior research [e.g., 16–18]. We used a 2 (own status = high or low) X 2 (others' status = high or low) between-subjects factorial design, in which we gave people feedback about their own and their fellow teammates' status in their team, and then assessed their subjective well-being. Participants were randomly assigned to condition using a randomizer in the computer-mediated survey flow which randomly assigned participants to one of four conditions evenly.

Five participants were recruited for each laboratory session. In Phase 1 of the study, participants completed a questionnaire at individual workstations, which included measures (e.g., personality traits and emotion recognition) that provided an ostensive basis for the status manipulation in Phase 2. Phase 2 presented the status manipulation. Participants were first told that other members of their group had also completed the same questionnaire they had just completed. They were provided (false) summaries of other members' personality and emotion-recognition scores, which ranged from high to low. All participants were shown the same set of false scores for their fellow teammates (see Appendix A in S1 Appendix). Participants were then told to rate, based on the scores they were provided, the status of each member of their group. They were told the aggregate status ratings of each group member would be used to determine which person would be selected as the leader of their group for a group exercise later in the study. Participants were not provided with their own scores and were led to believe that they and other group members only received other members' scores, but not self-scores. This helped avoid a potential confound–namely, a situation in which participants observed that their scores were better (or worse) than others, and thus reacted to the subsequent status feedback based on fairness concerns. Without any information about their own personality and emotion-recognition scores, participants would be less likely to believe that the status distributions were unfair to them.

After a brief wait in which each group member's status average status ratings were ostensibly calculated, participants were given false feedback regarding the status they and others were given (see Appendix B in S1 Appendix). Specifically, they were told that the range of possible status ratings was 1 to 7 and that their status (i.e. the median status ratings given to them by other members of their group) was either a 4 (in the own status = lower conditions) or a 6 (in the own status = higher conditions). They were also told that other group members' average status (i.e. the median status ratings that all the other members of their group received) was either a 4 (in the others' status = lower conditions) or 6 (in the other' status = higher conditions). We used a 1 to 7 scale because that is the most used scale in status research in psychology [e.g., 3, 29], and therefore our findings could be directly compared to prior studies of status.

In Phase 3, participants completed three measures of subjective well-being, and manipulation and suspicion checks. At the end of the study, they were debriefed and paid. Three participants expressed suspicion about the status feedback; no findings changed from significant to non-significant when we included or excluded those participants. We thus included them.

**Subjective well-being.**  Consistent with Anderson et al. (2012), we administered and combined the Positive and Negative Affect Schedule [30; for Positive Affect, $\alpha = 0.90$, $M = 4.44$, $SD = 1.05$; for Negative Affect, $\alpha = 0.88$, $M = 2.75$, $SD = 1.03$], and the Satisfaction with Life Scale, in which respondents provide a global, cognitive assessment of their life as a whole [31; SWLS; $\alpha = 0.81$, $M = 3.39$, $SD = 0.79$].

## Results and discussion

**Manipulation check.**  Participants were asked, "In your group, how much status do you have?" and "What about others in your group? On average, how much status do others have in your group" on a scale from 1 ("Very little") to 7 ("A lot"). The *own status* manipulation

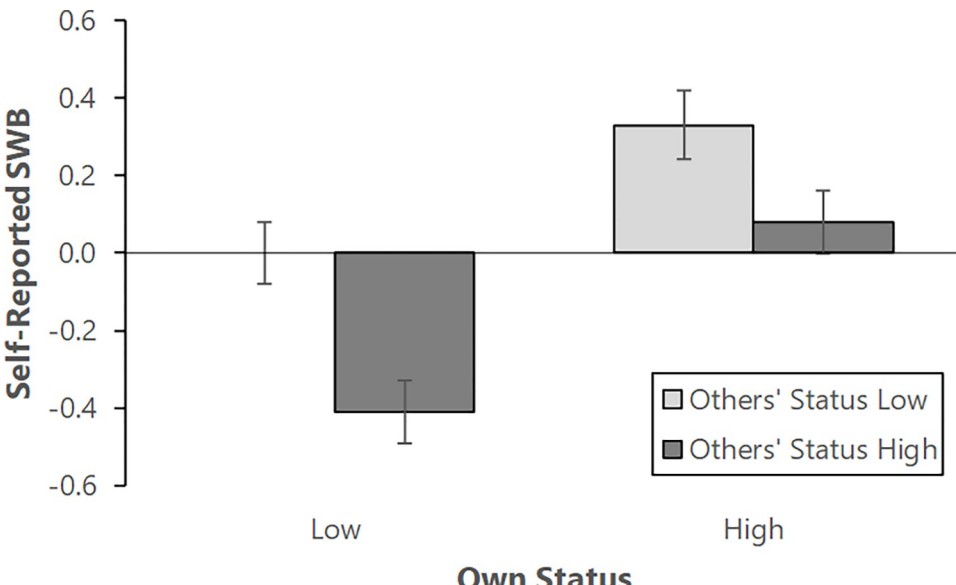

**Fig 1. Study 1: Average levels of subjective well-being broken down by condition.** Participants in the Low (/High) Own Status condition were told their own status was 4 (/6) out of 7. Participants in the Low (/High-) Others' Status condition were told the median status of other members of their group was 4 (/6) out of 7. After receiving the status feedback, participants rated their SWB. The figure shows mean SWB scores for participants in each condition. Error bars represent standard errors of the mean.

influenced participants' ratings of their own status ($M_{high}$ = 5.27, SD = 1.06; $M_{low}$ = 4.50, SD = 0.93) t(224) = 5.74, p < .001, d = 0.76, but not their ratings of others' status ($M_{high}$ = 4.77, SD = 0.88; $M_{low}$ = 4.65, SD = 0.96), t(224) = 0.94, p = .348, d = 0.13. The *others' status* manipulation influenced participants' ratings of others' status ($M_{high}$ = 5.21, SD = 0.93; $M_{low}$ = 4.21, SD = 0.57) t(224) = 9.80, p < .001, d = 1.30, but not ratings of their own status ($M_{high}$ = 4.89, SD = 1.09; $M_{low}$ = 4.88, SD = 1.04), t(224) = 0.01, p = .989, d = 0.00. There were no significant interactions. Therefore, the manipulations were effective.

**Hypothesis test.**   Fig 1 shows the effects of the different status distributions on subjective well-being. A two-way analysis of variance revealed a significant main effect for participants' own status, in that participants reported higher subjective well-being when their status was higher than when it was lower, F(1, 222) = 24.91, p < .001, $\eta^2$ = .101. This finding thus supports our hypothesis and conceptually replicates Anderson et al. (2012). There was also a significant main effect for others' status, in that participants had higher subjective well-being when others' status in the team was lower on average than when it was higher, F(1, 222) = 15.97, p < .001, $\eta^2$ = .067. There was not a significant interaction between participants' own and others' status, F(1, 222) = 1.01, p = .317, $\eta^2$ = .005.

It is interesting to note the simple effects. When participants' own status was lower, they had higher subjective well-being when other team members' status was also lower (M = .002, SD = .625) than when it was higher (M = -0.41, SD = 0.62), t(111) = 3.49, p = .001, d = 0.66. This finding is not particularly surprising, given the power of relative deprivation effects [31], and the dissatisfaction people feel when being worse off than others. However, even when participants' own status was higher, they had higher subjective well-being when other team members' status was lower (M = 0.33, SD = 0.64) than when it was higher (M = 0.08, SD = 0.57), t(111) = 2.15, p = .034, d = 0.40. Therefore, participants had higher subjective well-being when they alone had higher status than when they and their teammates all had higher (and equal) status (see Table 1).

**Table 1. Means and standard deviations for SWB in Studies 1 and 2.**

| Status Condition | | Study 1 | | Study 2 Control | | Study 2 Self-Affirm | |
|---|---|---|---|---|---|---|---|
| Own Status | Others' Status | *M* | *SD* | *M* | *SD* | *M* | *SD* |
| High | High | 0.08$_a$ | 0.57 | 0.08$_A$ | 0.51 | 0.34$_{BD}$ | 0.51 |
| High | Low | 0.33$_b$ | 0.64 | 0.14$_A$ | 0.66 | 0.18$_{AD}$ | 0.62 |
| Low | High | -0.41$_c$ | 0.62 | -0.42$_C$ | 0.64 | -0.39$_C$ | 0.73 |
| Low | Low | 0.00$_a$ | 0.63 | 0.04$_A$ | 0.61 | 0.02$_A$ | 0.63 |

Within each study, different subscripts imply means differ significantly from each other at the *p* < .05 level. Higher values imply higher SWB.

## Study 2

Is it possible to mitigate the effect of status on subjective well-being? Can any intervention cause people to feel better when their teammates have higher rather than lower status? To address this question, we examined whether asking participants to self-affirm would help mitigate the effects of status on subjective well-being, as prior research suggests that this intervention prompts individuals to be less interested in boosting their own status [27]. Therefore, Study 2 used the same design as in Study 1, but now included a self-affirmation manipulation for half of the participants. The methods and analysis plan for Study 2 were preregistered (https://aspredicted.org/LK8_DTB).

### Methods

**Participants.** Participants were 422 undergraduate students (66% female; average age = 20.80 years, *SD* = 3.08) who were paid $10.00 or given course credit. A power analysis based on the effect sizes observed in Study 1 suggested that at least 72 participants were needed to achieve 80% statistical power (two-tailed; $\alpha$ = .05) for a two condition design (288 for an eight condition design). Given sample size recommendations for replication studies, we increased the target sample size to 400 participants, consistent with the target sample size of roughly 50 participants per condition used in Study 1. Again, the participant pool was limited to undergraduate students and no other inclusion criteria were used. Participants were recruited between November 28, 2017, and April 6, 2018. All participants provided informed consent in writing which was verified by an experimenter prior to the start of the laboratory session and were then debriefed at the end of the study. Prior to the debriefing, participants were asked whether they found anything strange or unusual about the study, and one participant expressed suspicion about the status feedback. However, none of the conclusions or significant results reported below change after excluding this participant, so their data are included. The attention check questions used in Study 1 were used again in Study 2. Seventeen participants failed attention check questions or had technical issues and were thus excluded from the analyses, leaving 405 participants. Participants were asked to select all racial-ethnic categories to which they belonged; 47% selected Asian American, 27% selected White, 10% selected Latino, 0% selected African American, and 7% selected "other," and 8% declined to respond.

**Procedure and values affirmation.** The procedure, status manipulation, and measures were identical to Study 1, except we included a self-affirmation manipulation [25] right before participants received their status feedback. Those in the self-affirmation (/control) condition selected the three most important values (/least important values) from a list of 12 values that Cohen and colleagues [25] specified (membership in a social group [such as your community, racial group, or school club], relationships with friends or family, religious values, sense of

humor, living in the moment, music, creativity, being good at art, politics, athletic ability, being smart or getting good grades, and independence). They then described why these values are important to them (/might be important to someone else), listed the top two reasons why the values are important to them (/might be important to someone else), and rated the extent to which they agree or disagree with 4 statements related to the values, e.g. "These values have influenced me (/some people)." Study was therefore a 2 (own status = high vs. low) x 2 (average status of other members in the team = high vs. low) x 2 (intervention: self-affirmation vs. no affirmation) between-subjects factorial design. As per Study 1, participants were randomly assigned to condition using a randomizer in the computer-mediated survey flow which randomly assigned participants to one of the eight conditions evenly.

**Subjective well-being.** As in Study 1, we administered and combined the Positive and Negative Affect Schedule (for Positive Affect, $\alpha = 0.89$, $M = 4.49$, $SD = 1.02$; for Negative Affect, $\alpha = 0.90$, $M = 2.52$, $SD = 1.11$), and the Satisfaction with Life Scale ($\alpha = 0.80$, $M = 3.45$, $SD = 0.80$).

## Results and discussion

**Manipulation check.** The *own status* manipulation influenced participants' ratings of their own status ($M_{high} = 5.40$, $SD_{high} = 0.90$; $M_{low} = 4.25$, $SD_{low} = 0.96$) $t(400.8) = 12.45$, $p < .001$, $d = 1.24$, but not their ratings of others' status ($M_{high} = 4.84$, $SD_{high} = 0.93$; $M_{low} = 4.79$, $SD_{low} = 1.05$), $t(397.2) = 0.56$, $p = .575$, $d = 0.06$. The *others' status* manipulation influenced participants' ratings of others' status ($M_{high} = 5.40$, $SD_{high} = 0.93$; $M_{low} = 4.24$, $SD_{low} = 0.65$) $t(356.0) = 14.5$, $p < .001$, $d = 1.44$, but not ratings of their own status, $t(385.2) = -1.34$, $p = .181$, $d = -0.13$. There were no interactions. Furthermore, the self-affirmation manipulation did not affect participants' ratings of their own status($M_{Self-Affirm} = 4.82$, $SD_{Self-Affirm} = 1.08$; $M_{Control} = 4.84$, $SD_{Control} = 1.11$), $t(403) = -0.25$, $p = .803$, $d = -.03$, ratings of others' status ($M_{Self-Affirm} = 4.89$, $SD_{Self-Affirm} = 0.95$; $M_{Control} = 4.74$, $SD_{Control} = 1.03$), $t(403) = 1.53$, $p = .126$, $d = 0.15$, nor did it interact with the status manipulations in shaping participants' ratings of their own or others' status. Therefore, the status manipulations were effective in producing the effects and only the effects we intended, and the self-affirmation manipulation did not produce any unintended effects on participants' status ratings.

**Hypothesis tests.** Fig 2 shows the pattern of results, split into the control (Panel A) and self-affirmation (Panel B) conditions. As shown, in the control condition, participants again had higher subjective well-being when they had higher rather than lower status, $F(1, 200) = 12.63$, $p < .001$, $\eta^2 = .059$, as well as when others had lower rather than higher status $F(1, 200) = 9.74$, $p = .002$, $\eta^2 = .046$. Therefore, these findings replicate the results from Study 1.

In contrast to Study 1, however, there was a significant interaction between own and others' status, $F(1, 200) = 5.33$, $p = .022$ $\eta^2 = .026$. Fig 2 shows that this interaction is due to participants experiencing especially lower subjective well-being when their own status was lower, and their teammates' average status was higher (as opposed to when their own status was higher and their teammates' average status was higher). Again, this finding makes sense in light of relative deprivation research [32], which would suggest that being lower in status than others is particularly damaging to individuals' subjective well-being. In other words, when individuals possess higher status on an absolute level (6 out of 7), their subjective well-being is less damaged when others also have higher status on an absolute level; however, when individuals possess lower status on an absolute level (4 out of 7), their subjective well-being is particularly damaged when others have higher status on an absolute level. Being uniquely lower in status appears especially psychologically painful.

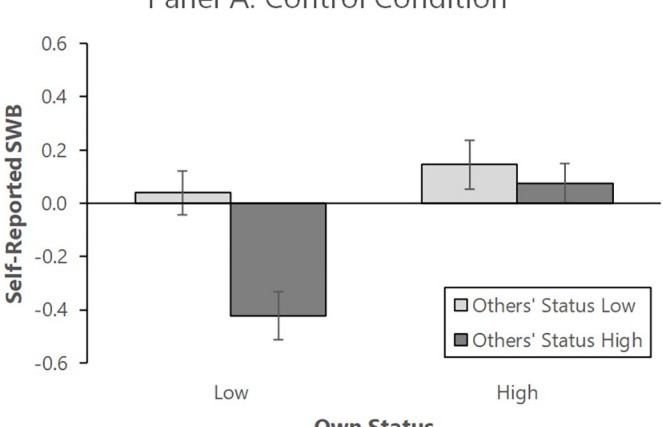

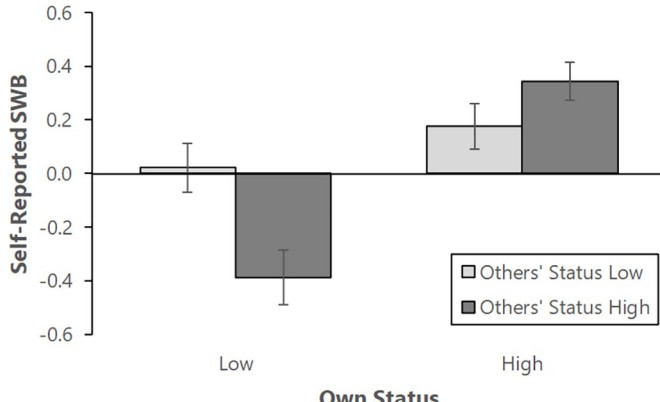

**Fig 2. Study 2: Average levels of reported subjective well-being broken down by condition.** (A) Participants in the Control Condition did not affirm their values before receiving feedback on their status. (B) Participants in the Self-Affirm condition affirmed their values before receiving feedback on their status. Participants in the Low (/High) Own Status condition were told their own status was 4 (/6) out of 7. Participants in the Low (/High-) Others' Status condition were told the median status of other members of their group was 4 (/6) out of 7. After receiving the status feedback participants rated their SWB. The figure shows mean SWB scores for participants in each condition. Error bars represent standard errors of the mean.

Turning to the self-affirmation manipulation, it did not moderate the effect of participants' own status on their subjective well-being, $F(1, 401) = 1.53$, $p = .217$, $\Delta R^2 = .004$, nor the effect of others' status on their subjective well-being, $F(1, 401) = 1.19$, $p = .276$, $\Delta R^2 = .003$. Therefore, the effects of own and others' status on subjective well-being was robust and largely unaffected by the self-affirmation manipulation. However, Fig 2 shows one condition that did differ between the control and self-affirmation conditions: Specifically, when participants' own status and their teammates' average status were all higher, they had higher subjective well-being in the self-affirmation condition, than in the control condition ($M_{Self-Affirm} = 0.34$, $SD_{Self-Affirm} = 0.51$; $M_{Control} = 0.08$, $SD_{Control} = 0.51$), $t(1, 98) = 2.64$, $p = .010$, $d = 0.53$. In the control condition, participants did not experience very high subjective well-being when everyone in their group was accorded higher status. However, in the self-affirmation condition, participants did experience higher subjective well-being when everyone in their group was accorded higher status (see Table 1). This lends at least some optimism that people's responses to status distributions can be changed.

## Discussion

Two studies, one preregistered, found that individuals' status had a causal effect on their subjective well-being. This finding provides a conceptual replication of Anderson et al. (2012), and suggests that the level of respect, admiration, and voluntary deference individuals are accorded by others does in fact shape their subjective well-being. That is, the correlations observed between status and subjective well-being in prior research might have stemmed from multiple causal forces. For example, being a happier person might have helped individuals achieve higher status, or third variables such as extraversion might have driven their effects (extraverts might tend to be happier and tend to achieve higher status). However, our findings suggest that the correlations observed are at the very least partly due to the causal effects of status.

Both studies also found that individuals' subjective well-being was not only shaped by their own status, but also others' status. Specifically, individuals had higher subjective well-being when others' status was lower than when it was higher. This suggests that people have a competitive orientation towards status; they not only want to have high status on an absolute level (e.g., to be highly respected and admired), but also to have higher status than others (e.g., to be more respected and admired than others). This idea has important implications. First, it potentially helps explain the pervasiveness of status stratification in groups. Status differences might pervade teams because people prefer and seek a form of stratification—specifically, a form in which they are higher than others—over egalitarianism. People's preferences might make egalitarianism highly unlikely. Second, this idea would suggest a somewhat provocative idea: that a fundamental human motive is substantially competitive in nature. Other fundamental motives, such as the needs for belongingness and autonomy, are not competitive, in that people do not seek to belong more than others or to be more autonomous than others [e.g., 33–35]. However, the desire for status might be unique among fundamental human motives in being competitive.

Study 2 asked whether self-affirmation might mitigate the effect of status on subjective well-being, and for the most part, the answer was "no." Self-affirmation has been shown to affect a range of thoughts, feelings, and behaviors [for a review, see 26]. However, it did not moderate the effect of individuals own' status, meaning individuals' own status shaped their subjective well-being, regardless of whether they self-affirmed or not. It also did not moderate the effect of others' status on subjective well-being, which indicates that individuals experienced higher subjective well-being when others' status was lower, even if they self-affirmed. The one finding that was affected by self-affirmation was the condition in which participants were told that they and their fellow group members had all been accorded higher status: in the control condition, individuals did not respond with high subjective well-being, whereas in the self-affirmation condition, individuals responded with higher subjective well-being. This suggests that at the very least, self-affirmation might help individuals be happier in groups where members are all afforded higher levels of respect and admiration.

However, overall, the idea that self-affirmation did not moderate the effects of status on subjective well-being warrants discussion. One possibility is that the effects of status on subjective well-being are so strong and robust that they are immune to interventions; perhaps nothing can reduce the psychological sting of possessing low status or dampen the pleasure of possessing high status. Another possibility is that the impact of status on subjective well-being can be moderated, but one must use other interventions besides self-affirmation to do so. For example, perspective-taking manipulations have been shown to increase empathy and concern for others [36]. It is possible that when individuals take the perspective of others and become more empathic towards them, they no longer experience higher subjective well-being when

those others have lower status, and perhaps even experience higher subjective well-being when those others have higher status.

These findings point to several directions for future research. First, that people had higher subjective well-being when others' status was lower suggests that people have a highly competitive orientation towards status—they feel better when they are above others. This contrasts with research on self-esteem [22], which suggested that people are relatively unaffected by their relative status position vis-à-vis others. One possibility worth examining is whether the effects of others' status on subjective well-being differ from the effects of others' status on self-esteem. Second, as mentioned above, given the ineffectiveness of self-affirmation as a possible way to mitigate the effects of status on subjective well-being, future research should continue seeking to identify other interventions that might be more effective. Third, we focused on main effects of status on subjective well-being, but do individual differences moderate these effects? For example, does status affect subjective well-being more strongly for narcissists, who care more about status than others, as compared to non-narcissists? Prior work suggests that individuals can base their self-esteem on different sources [37]. For example, some individuals might base their self-esteem on their achievements more than others. Inasmuch as subjective well-being is tied to self-esteem [9], this suggests an intriguing possibility: individuals might also differ in the extent to which they base their subjective well-being on different sources, including status. Some might derive their happiness from their status far more than others. Future research could examine individual differences in the degree to which individuals derive their subjective well-being from their status, and test whether these differences moderate the effect of status on SWB.

A potential limitation of the current studies is the reliance on undergraduate student populations and the potential generalizability of the findings to other subject populations, although we note that prior research on status and well-being suggests findings in this domain do appear to generalize [e.g., 3]. Related, the current research was conducted in a Western context, and it is possible that the observed effects might be stronger or weaker in different cultural contexts. A recent and comprehensive review of the research literatures on status [38] revealed that the psychology of status and specifically the desire for status appears to be universal, applying cross-culturally, but leaves open the possibility that the effects of status might be stronger in different contexts. Future research should examine this possibility. Another limitation is the potential for demand effects where participants may have guessed the hypotheses and be prompted to behave in a certain manner. The debriefing in both studies did not uncover this issue but future research should consider alternative non-computer-mediated designs to remove this possibility.

In summary, the current research provided the most direct evidence that individuals' status shapes their subjective well-being, found that individuals' subjective well-being was shaped not only by their own status but also by others', and that the effects of status on subjective well-being were largely immune to self-affirmation manipulations. These findings have several implication implications, including that individuals' standing in their group affects their well-being, that the human desire for status appears to be competitive in nature, and that the impact of status on subjective well-being is robust and not easily mitigated. Of course, given the research was conducted on undergraduate students in one country, and conducted in the laboratory, future research should test these effects on a broader range of individuals, in diverse countries, and ideally in contexts outside the laboratory. Nonetheless, the current findings take an important step in elucidating the impact that individuals' standing in their groups has on their well-being and psychological health.

## Supporting information

**S1 Appendix.**
(DOCX)

## Author Contributions

**Conceptualization:** Cameron Anderson.

**Data curation:** John Angus D. Hildreth.

**Formal analysis:** Cameron Anderson, John Angus D. Hildreth.

**Methodology:** Cameron Anderson.

**Project administration:** John Angus D. Hildreth.

**Writing – original draft:** Cameron Anderson.

**Writing – review & editing:** John Angus D. Hildreth.

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
