## [Decision Letter · Decision Letter 0]

1 May 2024

PONE-D-24-10935Status and subjective well-being: A conceptual replication and extension of Anderson et al. (2012)PLOS ONE

Dear Dr. Hildreth,

Thank you for submitting your manuscript to PLOS ONE. After careful consideration, we feel that it has merit but does not fully meet PLOS ONE’s publication criteria as it currently stands. Therefore, we invite you to submit a revised version of the manuscript that addresses the points raised during the review process.

 The revised version should address all comments. You may also note that unemployment is an important determinant of subjective well-being (https://doi.org/10.1002/hec.1361).

We look forward to receiving your revised manuscript.

Kind regards,

Petri Böckerman

Academic Editor

PLOS ONE

Journal Requirements:

Additional Editor Comments:

The revised version should address all comments.

Comments from PLOS Editorial Office:

We note that one or more reviewers has recommended that you cite specific previously published works. As always, we recommend that you please review and evaluate the requested works to determine whether they are relevant and should be cited. It is not a requirement to cite these works. We appreciate your attention to this request.

Reviewers' comments:

Reviewer's Responses to Questions

**Comments to the Author**

1. Is the manuscript technically sound, and do the data support the conclusions?

Reviewer #1: Partly

Reviewer #2: Yes

2. Has the statistical analysis been performed appropriately and rigorously? 

Reviewer #1: Yes

Reviewer #2: Yes

3. Have the authors made all data underlying the findings in their manuscript fully available?

Reviewer #1: Yes

Reviewer #2: Yes

4. Is the manuscript presented in an intelligible fashion and written in standard English?

Reviewer #1: Yes

Reviewer #2: Yes

5. Review Comments to the Author

Reviewer #1: The topic is fascinating, but a thorough restructuring of the content is necessary. I kindly request the esteemed author to revise the entire article with a deeper and more coherent perspective, taking into consideration the suggested changes. Afterward, please resubmit it for further review. I believe this feedback will be valuable in improving the quality of your work.

Best regards.

Abstract

1. The could benefit from a clearer structure. Consider breaking it down into distinct sections, such as Introduction, Methods, Results, and Conclusion.

2. Begin with a concise introduction to provide context and importance of the research question.

3. Provide more details about the experiments conducted, including sample size, demographics, and any relevant variables controlled for.

4. Expand on the implications of the findings for theory and practice.

introduction

• Consider providing a bit more context on the theoretical frameworks that underpin the relationship between status and SWB, such as social comparison theory or self-esteem theory.

• The introduction clearly outlines the three primary aims of the current research, including a conceptual replication of previous findings, examination of the effect of others' status on SWB, and exploration of potential interventions.

• To enhance the quality of your introduction, consider incorporating the following references:

https://link.springer.com/article/10.1007/s10943-020-01151-z

http://jpcp.uswr.ac.ir/browse.php?a_id=787&sid=1&slc_lang=fa&ftxt=0

• Conclude the introduction by summarizing the main points discussed and clearly transitioning to the subsequent sections of the paper, such as the methodology and results.

• Ensure the introduction sets the stage effectively for the reader and generates interest in the research questions addressed.

Study 1

1. It's important to clarify whether the participant pool was limited to undergraduate students or if it included individuals from other backgrounds.

2. Specify if there were any inclusion criteria other than being an undergraduate student (e.g., age range, academic major) to provide a clearer picture of the sample.

3. Ensure clarity in the description of the manipulation process. For instance, it's mentioned that participants were provided with false feedback about their status and others', but it's not entirely clear how this feedback was presented and if there were any variations in presentation among conditions.

4. Clarify the rationale behind using a 1 to 7 scale for status ratings and ensure that this choice aligns well with the manipulation and the constructs under investigation.

5. Provide details on the selection and construction of attention check questions to ensure they effectively measure participants' engagement and attentiveness.

6. Provide effect sizes (e.g., Cohen's d) for all significant findings to supplement the significance testing and enhance the interpretation of results.

7. Consider discussing any potential limitations of the study, such as the generalizability of findings beyond the undergraduate student population or the potential impact of demand characteristics on participants' responses.

8. Provide detailed descriptions of the figures and tables in the text to guide readers in understanding the key findings presented in visual format.

9. Ensure that the study adheres to ethical guidelines, particularly regarding informed consent, debriefing procedures, and handling of participant data.

Study 2

• Provide more detailed information about the rationale behind the sample size selection and power analysis. Justify why 400 participants were targeted for the study.

• Consider providing more information on the demographic characteristics of the participants beyond just race and gender. Factors such as socioeconomic status, cultural background, and education level may influence subjective well-being and could be relevant to the study.

• it would be beneficial to include a rationale for choosing the specific values included in the values affirmation manipulation. Justify why these values were selected and how they relate to the study's objectives.

• Provide more details on the randomization procedure to ensure that participants were evenly distributed across conditions.

• Ensure clarity in the presentation of results. Some sentences are convoluted and could be simplified for better comprehension.

• When reporting statistical results, include effect sizes along with p-values to provide a more comprehensive understanding of the findings.

• Consider including a table summarizing the main results to aid readers in quickly understanding the key findings.

• Provide more detailed interpretations of the results, particularly regarding the implications of the interaction effects observed in the study.

• Propose avenues for future research based on the study's findings, including potential follow-up experiments or longitudinal studies to further explore the effectiveness of interventions on subjective well-being.

discussion

1. there are some instances where the phrasing could be clarified for better understanding. For example, in the sentence "That is, while the correlations observed between status and SWB in prior research might have stemmed from multiple causal forces," it could be clearer if you specify what these multiple causal forces might be.

2. When discussing Study 2, where the authors explored the potential moderating effect of self-affirmation on the relationship between status and SWB, it would be beneficial to clearly delineate the main findings. Make it explicit which aspects of the relationship were affected by self-affirmation and which were not.

3. it could be beneficial to delve deeper into potential alternative explanations. For instance, apart from the proposed causal effect of status on SWB, are there other variables or mechanisms that could explain the observed relationship? Discussing potential alternative explanations would strengthen the robustness of the findings.

4. Provide more detailed suggestions for future studies. For example, in addition to examining individual differences in the relationship between status and SWB, what specific aspects of personality or social context might moderate this relationship? Providing concrete research questions and hypotheses for future studies would enrich the discussion.

5. Discuss how cultural differences might influence the relationship between status and SWB. Are there cultural norms or values that could affect how individuals perceive and respond to status? Considering cultural variations would provide a more comprehensive understanding of the findings and their generalizability.

6. Conclude the discussion by summarizing the key findings, implications, and avenues for future research. Reinforce the significance of the findings in advancing understanding of the relationship between status and SWB, while also acknowledging limitations and areas for further exploration.

Reviewer #2: The discussion section of the article feels somewhat abrupt and lacks depth in its analysis. The discussion transitions quickly between different ideas without providing a thorough exploration of each point. For instance, while the competitive orientation towards status is briefly mentioned, further elaboration on its theoretical implications and practical relevance is warranted. Additionally, the finding that self-affirmation does not significantly moderate the effect of status on SWB requires more nuanced interpretation and consideration of alternative explanations.

Also, I question the suitability of undergraduate students as a sample population for examining the hypothesis addressed in this study. Typically in their twenties and yet to embark on their professional careers, undergraduate students may not undergo significant status changes in the short term, potentially limiting the generalizability of findings. Moreover, it appears that the study overlooks addressing the limitation of generalizing conclusions drawn from the experiments.

6. PLOS authors have the option to publish the peer review history of their article (what does this mean?). If published, this will include your full peer review and any attached files.

Reviewer #1: No

Reviewer #2: No

---

## [Author Response · Author response to Decision Letter 0]

13 May 2024

Responses to reviewer comments have been made in the response to reviewer comments note attached. For reference the comments have been copied below.

Response to Reviewers

[Original text shown indented in italics]

PONE-D-24-10935

Status and subjective well-being: A conceptual replication and extension of Anderson et al. (2012)

PLOS ONE

Dear [ANONYMIZED],

Thank you for submitting your manuscript to PLOS ONE. After careful consideration, we feel that it has merit but does not fully meet PLOS ONE’s publication criteria as it currently stands. Therefore, we invite you to submit a revised version of the manuscript that addresses the points raised during the review process.

The revised version should address all comments. You may also note that unemployment is an important determinant of subjective well-being (https://doi.org/10.1002/hec.1361).

We look forward to receiving your revised manuscript.

Kind regards,

Petri Böckerman

Academic Editor

PLOS ONE

Journal Requirements:

RESPONSE: Thank you for your and the review team’s comments and advice and for the opportunity to submit a revised manuscript. As requested we have included this Response to Reviewers as well as a marker-up and unmarked version of the revised manuscript. We confirm the following:

• We don’t wish to submit an updated financial disclosure. 

• We have resubmitted our figure files in tif format per the guidelines. 

• The lab protocols have been submitted to OSF (where the data had been stored) rather than protocols.io to reduce the burden on readers having to navigate between different systems.

• We have ensured the revision meets PLOSOne’s formatting guidelines. Thank you for sharing the templates.

• We have included the full ethics statement in the Methods sections of Study 1. Specifically, we have edited the prior statement to now read: “The research in both studies was conducted in accordance with the principles expressed in the Declaration of Helsinki and the protocol was approved by The Committee for Protection of Human Subjects comprising the Institutional Review Boards of the University of California, Berkeley (IRB Approval ID: 2015-01-7044). All subjects gave their informed consent for inclusion before they participated in the study. and were then debriefed at the end of the study.” 

• We also included the citation relating to the relationship between unemployment and subjective health-assessments in the revised manuscript (Bockerman & Ilmakunnas, 2009).

Additional Editor Comments:

The revised version should address all comments.

Comments from PLOS Editorial Office:

We note that one or more reviewers has recommended that you cite specific previously published works. As always, we recommend that you please review and evaluate the requested works to determine whether they are relevant and should be cited. It is not a requirement to cite these works. We appreciate your attention to this request.

Reviewers' comments: Thank you for clarifying. 

Reviewer's Responses to Questions

Comments to the Author

1. Is the manuscript technically sound, and do the data support the conclusions?

Reviewer #1: Partly

Reviewer #2: Yes

2. Has the statistical analysis been performed appropriately and rigorously? 

Reviewer #1: Yes

Reviewer #2: Yes

3. Have the authors made all data underlying the findings in their manuscript fully available?

Reviewer #1: Yes

Reviewer #2: Yes

4. Is the manuscript presented in an intelligible fashion and written in standard English?

Reviewer #1: Yes

Reviewer #2: Yes

5. Review Comments to the Author

Reviewer 1 comments

Reviewer #1: The topic is fascinating, but a thorough restructuring of the content is necessary. I kindly request the esteemed author to revise the entire article with a deeper and more coherent perspective, taking into consideration the suggested changes. Afterward, please resubmit it for further review. I believe this feedback will be valuable in improving the quality of your work.

Best regards.

RESPONSE: We have made substantial revisions to the paper, throughout all sections, based on your comments, the other reviewer’s comments, and the editor’s suggestions. We discuss below how we addressed each of your comments in turn. 

Abstract

1. The could benefit from a clearer structure. Consider breaking it down into distinct sections, such as Introduction, Methods, Results, and Conclusion.

2. Begin with a concise introduction to provide context and importance of the research question.

3. Provide more details about the experiments conducted, including sample size, demographics, and any relevant variables controlled for.

4. Expand on the implications of the findings for theory and practice.

RESPONSE: Thank you for this suggestion. We examined the abstracts of several articles published in PLOS ONE and found that most of them used the same structure in the abstract that we did; fewer used the breakdown you suggested. Therefore, in keeping with what appears to be more of the norm in PLOS ONE articles, we retained the original structure of our abstract.

introduction

• Consider providing a bit more context on the theoretical frameworks that underpin the relationship between status and SWB, such as social comparison theory or self-esteem theory.

• The introduction clearly outlines the three primary aims of the current research, including a conceptual replication of previous findings, examination of the effect of others' status on SWB, and exploration of potential interventions.

• To enhance the quality of your introduction, consider incorporating the following references:

https://link.springer.com/article/10.1007/s10943-020-01151-z

http://jpcp.uswr.ac.ir/browse.php?a_id=787&sid=1&slc_lang=fa&ftxt=0

• Conclude the introduction by summarizing the main points discussed and clearly transitioning to the subsequent sections of the paper, such as the methodology and results.

• Ensure the introduction sets the stage effectively for the reader and generates interest in the research questions addressed.

RESPONSE: Based on your suggestions, we elaborated on why status might shape SWB, particularly the possibility that status shapes self-esteem, which in turn influences SWB. We also included a statement at the end of the introduction that summarized its main points, which helped transition to the “roadmap” that outlined the two studies we conducted. We considered incorporating the two references you mentioned, but upon reading them, failed to grasp their relevance to the current research.

Study 1

1. It's important to clarify whether the participant pool was limited to undergraduate students or if it included individuals from other backgrounds.

RESPONSE: In the methods sections of both studies, we specify that “Participants were XXX undergraduate students…” To remove any doubt of the subject population we have clarified that the participant pool in each study “was limited to undergraduate students.”

2. Specify if there were any inclusion criteria other than being an undergraduate student (e.g., age range, academic major) to provide a clearer picture of the sample.

RESPONSE: We have clarified in the methods section to each study of the revised manuscript that “No other inclusion criteria were used.”

3. Ensure clarity in the description of the manipulation process. For instance, it's mentioned that participants were provided with false feedback about their status and others', but it's not entirely clear how this feedback was presented and if there were any variations in presentation among conditions.

RESPONSE: To clarify the status manipulation process, we have added two appendices in the revised manuscript with the exact details of the process and have clarified what information was presented to all participants and what information was provided just to those in specific conditions.

4. Clarify the rationale behind using a 1 to 7 scale for status ratings and ensure that this choice aligns well with the manipulation and the constructs under investigation.

RESPONSE: In the methods section of Study 1, we have further clarified that “We used a 1 to 7 scale because that is the most used scale in status research in psychology (e.g., Anderson et al., 2001; Cheng et al., 2013).” The definition of status used in the manipulations “That is, how much should each member be respected and admired in the group, how much should they lead the group’s task, and how much should others voluntarily defer to them?” (see Appendices), is consistent with the definition of status used in the introduction (second sentence): “Status is the respect, admiration, and voluntary deference individuals are afforded by others.”

5. Provide details on the selection and construction of attention check questions to ensure they effectively measure participants' engagement and attentiveness.

RESPONSE: In the revised manuscript, we have clarified what attention checks were used to ensure participants were attentive: “During the initial questionnaire, described below, two attention check questions were included that asked participants to select specific responses, i.e., “If I am paying attention to this study, I will select the agree a little option,” and “If I am paying attention I will select this answer. /I will not select this answer. This method of detecting inattentive participants has been used extensively in past research to increase the quality of data (e.g., Shamon & Berning, 2020).” We note that attention checks of this nature are very common in psychological research (e.g., Huan & Bowling, 2015; Kam & Meyer, 2015; Meade & Craig, 2012; Shamon & Berning, 2020).

Huang JL, Bowling NA, Liu M, Li Y. Detecting insufficient effort responding with an infrequency scale: Evaluating validity and participant reactions. Journal of Business and Psychology. 2015;30:299-311.

Kam CCS, Meyer JP How careless responding and acquiescence response bias can influence construct dimensionality: The case of job satisfaction. Organizational research methods. 2015;18(3):512-541.

Meade AW, Craig SB. Identifying careless responses in survey data. Psychological methods. 2012;17(3):437.

Shamon H, Berning CC. Attention Check Items and Instructions in Online Surveys with Incentivized and Non-Incentivized Samples: Boon or Bane for Data Quality?. Survey Research Methods. 2020;14(1):55-77. 

6. Provide effect sizes (e.g., Cohen's d) for all significant findings to supplement the significance testing and enhance the interpretation of results.

RESPONSE: Thank you for highlighting. Effect sizes (Cohen’s d and partial eta squared) have been added to all significant results in both studies in the revised manuscript.

7. Consider discussing any potential limitations of the study, such as the generalizability of findings beyond the undergraduate student population or the potential impact of demand characteristics on participants' responses.

RESPONSE: We have added a limitation section to the General Discussion rather than to each study which highlights these potential issues. Specifically, we comment “A potential limitation of the current studies is the reliance on undergraduate student populations and the potential generalizability of the findings to other subject populations, although we note that prior research on status and well-being suggests findings in this domain do appear to generalize (e.g., Anderson et al., 2012). Another limitation is the potential for demand effects where participants may have guessed the hypotheses and be prompted to behave in a certain manner. The debriefing in both studies did not uncover this issue but future research should consider alternative non-computer mediated designs to remove this possibility.”

8. Provide detailed descriptions of the figures and tables in the text to guide readers in understanding the key findings presented in visual format.

RESPONSE: Legends have been ad

---

## [Editor Report · Decision Letter 1]

7 Aug 2024

Status and subjective well-being: A conceptual replication and extension of Anderson et al. (2012)

PONE-D-24-10935R1

Dear Dr. Hildreth,

We’re pleased to inform you that your manuscript has been judged scientifically suitable for publication and will be formally accepted for publication once it meets all outstanding technical requirements.

Kind regards,

Petri Böckerman

Academic Editor

PLOS ONE

Additional Editor Comments (optional):

I am happy with the revised paper and responses.
---

## [Editor Report · Acceptance letter]

23 Aug 2024

PONE-D-24-10935R1 

PLOS ONE

Dear Dr. Hildreth, 

I'm pleased to inform you that your manuscript has been deemed suitable for publication in PLOS ONE. Congratulations! Your manuscript is now being handed over to our production team.

Kind regards, 

on behalf of

Professor Petri Böckerman 

Academic Editor

PLOS ONE